# Comparative Analysis of Vegetable and Mineral Oil-Based Antiadhesive/Hydrophobic Liquids and Their Impact on Wood Properties

**DOI:** 10.3390/ma16144975

**Published:** 2023-07-12

**Authors:** Magdalena Kachel, Anna Krawczuk, Marta Krajewska, Stanisław Parafiniuk, Tomasz Guz, Klaudia Rząd, Arkadiusz Matwijczuk

**Affiliations:** 1Department of Machinery Exploitation and Management of Production Processes, Faculty of Production Engineering, University of Life Sciences in Lublin, 28 Głęboka St., 20-612 Lublin, Poland; magdalena.kachel@up.lublin.pl (M.K.); anna.krawczuk@up.lublin.pl (A.K.); stanislaw.parafiniuk@up.lublin.pl (S.P.); 2Department of Biological Bases of Food and Feed Technologies, Faculty of Production Engineering, University of Life Sciences in Lublin, 20-612 Lublin, Poland; 3Department of Food Engineering and Machinery, University of Life Sciences in Lublin, 20-612 Lublin, Poland; tomasz.guz@up.lublin.pl; 4Department of Biophysics, University of Life Sciences in Lublin, Akademicka 13, 20-950 Lublin, Poland; k.terlecka.98@wp.pl (K.R.); arkadiusz.matwijczuk@up.lublin.pl (A.M.); 5ECOTECH-COMPLEX—Analytical and Programme Centre for Advanced Environmentally-Friendly Technologies, Maria Curie-Sklodowska University, Głęboka 39, 20-033 Lublin, Poland

**Keywords:** biodegradation, vegetable oils, mineral oils, hydrophobic liquids, raw wood, FTIR spectroscopy

## Abstract

The unavailability of biodegradable preservatives is one of the major setbacks in the construction industry. With this in mind, our study focused on the analysis and comparison of two hydrophobic liquids, one vegetable oil-based (VOA) and the other mineral oil-based (MOA), and subsequently applying the same on three types of wood. The comparison of the vegetable oil-based (VOA) and mineral oil-based (MOA) *hydrophobic liquids* revealed that VOA was characterized by an 83.4% susceptibility to aerobic biodegradation, while MOA was considerably more resistant (47.80%). Based on the conducted contact angle measurements, it was observed that the wettability of pine and oak wood decreased after the application of both VOA (for pine—twice; for oak—by 38%) and MOA (for pine—more than two times; for oak—by 49%), while in the case of aspen, the same was increased (after the application of VOA—by 20%; after the application of MOA—by 2%). The observed depth of penetration into the structure of the impregnated wood was lower for the VOA impregnant as compared to the MOA impregnant. This result persisted in all types of wood used in the experiment. Observations of the process of water absorption during soaking revealed that VOA was more beneficial in terms of lowering water absorption into the material, regardless of wood type. The overall results were better for VOA, which lowered the mass of soaked wood by between 19.73 and 66.90%.

## 1. Introduction

Hydrophobic liquids are commonly used to protect construction materials such as concrete or wood against harmful external factors. Maintaining construction materials and structures in good condition by ensuring suitable protection against sunlight and humidity, which can cause swelling, cracking, and fungal attacks, benefits sustainable economic development and helps to eliminate excessive emissions of harmful substances into the environment. Given the lack of natural agents that could provide such protection, many researchers as well as producers are currently actively investigating this problem. The research focuses, in particular, on developing industrial chemicals that would be more sustainable and would facilitate a better quality, affordability, and environmental friendliness of biodegradable products that can be obtained from renewable sources (e.g., vegetable oil) [1,2].

As we know, natural products are biodegradable. This process can be significantly accelerated by employing compounds that are naturally more vulnerable to decomposition. The same include, e.g., polymers for which the main chains contain groups susceptible to hydrolytic microorganism attacks, i.e., ester, carboxyl, hydroxyl, or ether groups [3]. Wood is a renewable material widely used in a variety of applications, including construction. Given its ubiquity, the development of more efficient methods of wood preservation is crucial. The quality and durability of wood material are negatively affected by numerous external factors, such as fungal attacks [4,5], surface erosion, or dimensional changes [6], all of which are related to wood’s hydrophilic character and its resulting susceptibility to interactions with water. The high hydrophilic absorbability of a biopolymer negatively impacts its dimensional stability and resistance to decomposition [7,8]. Wood is predominantly composed of cellulose, hemicellulose, and lignin, all of which are naturally hydrophilic due to the presence of hydroxy groups in their molecules [9]. Numerous studies have been undertaken with a view to find new chemical, biological, or physical methods of treating wood surfaces to improve their properties [10,11,12,13]. Currently, the most common agents used to protect natural construction materials are of a petrochemical origin. However, one of our priorities should be to develop coating materials that would complement wood’s inherent renewability and low environmental impact while preserving its natural ability to buffer moisture.

Hydrophobic liquids used for such purposes must meet a number of requirements in terms of, e.g., facilitating the clean separation of, for example, boarding from hardened concrete during demolition, or the ability to provide long-term protection. The effectiveness of the preservation and external protection of wood depends largely on the depth to which the given impregnant can penetrate into the material, which is why wood should only be impregnated when dry [13].

When it comes to protecting wooden structures, most of the agents currently available on the market are refined compounds, paraffin-based mineral oils, or oil emulsions [14,15]. In recent years, a number of researchers have been looking into the possibility of developing super-hydrophobic coatings that could greatly improve the water resistance of raw wood [16,17]. As follows, from a study conducted by Syahrullail et al. in 2011 [18], vegetable oils were found to be superior to mineral oils in terms of resistance to wear due to abrasion and fatigue, lower toxicity, and better biodegradability, but at the same time showed worse thermal and oxidative stability [19,20]. Their hydrophobic properties allow them to be used in the production of hydrophobic agents [21], and consequently also in hydrophobizing formulations [20].

Given the fact that the currently available literature lacks reports on the uses of vegetable oils as hydrophobic agents protecting wooden surfaces, we decided to take up this challenge and conduct this study with a view to compare the effectiveness of two hydrophobic compounds (VOA and MOA) applied as protective coatings onto three types of wood.

The research problem pertained to the question of how the two hydrophobic compounds influenced the quality of the coated material. The respective hydrophobic liquids were analysed in terms of aerobic biodegradability, viscosity, and density. Surface tension was also measured, as well as the contact angles when applying distilled water on the surfaces of the wood samples. The water was applied prior to and after the absorption and drying (drying time 24 h) of the two impregnants. Both liquids also underwent qualitative analyses with the use of FTIR spectroscopy. Dried wood samples, untreated and coated with the hydrophobic liquids, were tested for their moisture content, density, and absorbability. The depth to which the agents penetrated into the wood was also measured.

## 2. Materials and Methods

### 2.1. Origin of the Research Material

The research material consisted of two hydrophobic liquids, one containing industrial rapeseed oil (VOA) and the other a mineral oil-based formulation commercially available on the domestic market (MOA), as well as three types of raw wood samples: aspen, pine, and oak. The pine (*Pinus sylvestris* L.), aspen (*Populus tremula* L.), and oak (*Quercus* L.) woods were obtained from a mixed forest located in the eastern part of Lubelskie Voivodeship in Poland, 51.686923, 23.295797, Podedwórze. The trees had been felled in the winter of 2018 (March), after which their trunks were planked and seasoned outdoors with wooden strip spacers in place to ensure uninterrupted air circulation. The timber piles were covered from above to protect them against precipitation. The wood was seasoned outdoors for a period of one year. Afterwards, the planks were stored indoors, in a properly aired space, with adequate spacers in place. The wood was then used to prepare samples for the study. Selected wood pieces were cut to size (1 m by 30 cm) in triplicate.

The hydrophobic liquids (VOA and MOA) were applied on the wood surface by submerging the samples in a shallow tray filled with the liquid in question (5 L). The wood samples were cut to 1 m × 30 cm. The soaking time was 30 s. Afterwards, the samples were dried in laboratory conditions for 24 h at 20 ± 2 °C.

### 2.2. Measurement of Liquid Density and Viscosity

Density was measured with the hydrostatic method using a Radwag analytic scale XA 110.

The hydrophobic liquid was placed in a container (beaker) and a plunger was placed in the fluid. The plunger was completely submerged. The fluid density ρ was calculated as the ratio of the mass of the fluid m to its volume V, measured in constant temperature as per the following formula:(1)$$\rho =\frac{m}{v}$$

Kinematic viscosity at 20 °C was measured in accordance with the EN ISO 3104 standard [22] using certified Ubbelohd capillary tubes (Size 1B: measurement range 10–50 mm^2^/s c 0.05; Size 2B: measurement range 100–500 mm^2^/s c 0.5; Size 3: measurement range 200–1000 mm^2^/s, c1). PSL-Rheotek baths were used. The dynamic viscosity was calculated using the following formula:η = ν·ρ(2)
where

η—dynamic viscosity (mPa);

ν—kinematic viscosity (mm^2^/s);

ρ—density (g/cm^3^).

The measurements were conducted in triplicate to verify that the discrepancies between the respective results (repeatability) did not exceed ±0.5%.

### 2.3. Biodegradability Testing

Tests with respect to aerobic biodegradability in aqueous media were performed with the use of manometric respirometry. The samples were incubated for 28 days, with access to oxygen, at 22 ± 1 °C, with continuous stirring, and inside tightly sealed respirometry containers. The concentration of the analysed material was 100 mg/L, i.e., the sample contained at least 50–100 mg of ThOD/L with the adequate amount of inoculum, up to 30 mg d.m./L, and mineral medium. The active sediment for the analyses was collected from an aeration tank at the “Czajka” Water Treatment Plant in Warsaw. The degree of biodegradability was determined by measuring oxygen consumption and was expressed directly as mg/L BOD. The amount of oxygen absorbed by the sediment microorganisms during the biodegradation of the analysed material was expressed as a ThOD or COD percentage. The tests were consistent with appendices 1 and 3 to SPO/BS/01/b (rev. 5) and the PN ISO 9408:2005 standard [23].

### 2.4. Measurements of Surface Tension and Contact Angle

The measurements of the surface tension of the hydrophobic liquids (MOA and VOA) and the contact angles during the application of distilled water onto wood surfaced after the absorption and drying (drying time 24 h) of the two impregnants, were conducted using a DSA30 KRÜSS goniometer. Surface tension was measured with the pendant drop method. A drop of the hydrophobic liquid was dosed with a weight that allowed it to remain suspended under the tip of the dosing needle. The shape of the drop was then analysed to calculate the surface tension at the liquid–air interface. A total of 30 measurements were conducted for each of the hydrophobic liquids analysed. For comparison, 30 surface tension measurements were also taken for distilled water.

Contact angle measurements were performed by evaluating and analysing digital images of drops of distilled water applied onto wood surfaced after the absorption and drying (drying time 24 h) of the two hydrophobic liquids (MOA and VOA). The impregnated wood sample was placed on the measurement table and an automatic NICHIRYO Le-20 pipette was used to apply a 4 µL drop of the distilled water onto the material. The tests were performed using the method described as the “Tangent Method-1” by the device’s manufacturer, which facilitates the measurement of the contact angle, i.e., the angle between the surface of the solid and the tangent of the liquid meniscus curve from the point of contact of the three phases: solid, liquid, and gaseous. The contact angles were measured after impregnating the wood, once the impregnant was fully absorbed and dried (drying time 24 h), with two agents (a vegetable oil-based hydrophobic liquid and mineral oil-based hydrophobic liquid). Ten contact angle measurements were taken for each of the wood samples (pine, aspen, oak) after the application of distilled water. All surface tension and contact angle measurements were performed at 20 ± 1°C.

### 2.5. Measurement of Wood Density

The tests were conducted in accordance with the following standards:

PN-EN 384 + A1:2018-12 [24]. Structural timber. Determination of characteristic values of mechanical properties and density. PN-77/D-04100 [25]. Timber. Density measurement.

The density for the initial water content (at the given moment) [W] was calculated with 5 kg/m^3^ accuracy (0.005 g/cm^3^) using the following formula:(3)$$\rho_{w}=\frac{m_{w}}{a_{w}\cdot b_{w}\cdot l_{w}}=\frac{m_{w}}{V_{w}}$$
where

$m_{w}$—sample mass for the given water content W [g];

$a_{w},b_{w},l_{w}$—sample dimensions [mm];

$V_{w}$—sample volume [mm^3^].

The obtained value was thus normalised for 12% water content, and, if 7 ≤ W ≤ 17, the following formula was used:(4)$$P_{w}=\rho_{w}\cdot \left[1-\frac{\left(1-K\right)\cdot \left(W-12\right)}{100}\right]$$
where

K—volumetric shrinkage coefficient;

W—mean wood water content [%].

### 2.6. Measurement of Water Content in Wood

The water content in wood was measured in accordance with the following standard: PN-EN 408 + A1:2012 [26]. Timber structures. Structural timber and glued laminated timber. Determination of some physical and mechanical properties. The wood samples were weighed raw and dried at 103 ± 2 °C (apart from pine, which, due to its high resin content, was dried at 50 °C), until the discrepancy between two subsequent weight measurements was less than 0.1%. The measurements were taken every two hours.

### 2.7. Measurement of Wood Absorbability

Wood absorbability was measured in accordance with the following standards: PN-59/D-0449 [27]. Physical and mechanical properties of timber. Absorbability measurement. PN-77/D-04227 [28]. General principles of sampling and sample preparation, and PN-77/D-04100 [25]. Measurement of water content.

The test material was dried at 103 ± 2 °C to reach 0% water content, and then submerged in distilled water in a covered container at 20 ± 2 °C. The samples were removed from the container every 6 h, dried with blotting paper, and weighed until the discrepancy between the subsequent measurements did not exceed 0.01g. The rate of absorption was calculated using the following formula:(5)$$V_{n}=\frac{W}{t}(\frac{\%}{h})$$
where

W—sample water content [%];

$V_{n}$—rate of absorption [%/h];

t—time of absorption [h].

### 2.8. Fourier Transform Infrared Spectroscopy—FTIR

The ATR spectra for the surface layer of the analysed samples and the solutions were registered with the use of an ATR attachment with a diamond crystal. The ART spectra were registered on an FTIR Thermo Nicolet 8700 spectrometer with a Smart Orbit™ diamond ATR attachment and DTGS (Deuterated Triglycine Sulphate) detector. The spectra underwent ATR correction, baseline correction, and ATR scale normalisation to achieve transmission spectra equivalence.

The quantitative measure of the quality of fit between the measured spectrum and the benchmark is the correlation index or HQI (hit quality index), where HQI = 100 indicates a perfect spectral fit. Values of HQI > 90 indicate very high levels of spectral similarity. The results of the FTIR-ATR analyses are presented in a graphic format as sets of comparative spectra. The attachment crystal was always thoroughly cleaned using ultrapure solvents purchased from Sigma-Aldrich (Saint Louis, MO, USA). The spectra were measured within the spectral range from 450 to 3700 cm^−1^ at a resolution of 2 cm^−1^. The results were processed and prepared for publication using Grams AI software from ThermoGalactic Industries (Waltham, MA, USA) and OriginPro 2021 software from OriginLab Corporation (Northampton, MA, USA).

### 2.9. Measurement of the Impregnation Depth

The analysed wood samples (oak, aspen, pine) were treated with two types of impregnants: MOA and VOA. Both impregnants were applied on the wood surface in close proximity using a brush. To avoid the mixing of the antiadhesive/hydrophobic liquids, the application areas were separated with insulation tape. The liquids were stained with an intense violet.

After the absorption and drying of the antiadhesive/hydrophobic fluid (drying time 24 h), the samples were precision-cut perpendicularly to the impregnated surface. The surface of the cut was smoothed to facilitate microscopic observation. A PFM Slide 2002 sliding microtome was used for this purpose. Once the knife edge was aligned, multiple runs were performed by moving the sample in 5 µm increments until smooth. Afterward, the microtone knife was replaced, and the process was repeated several times by removing layers of the material in 2 µm increments.

When prepared, the samples were analysed on an Olympus CX-41 optical microscope. Once the impregnant penetration layer was identified, the observed images were photographed with an Olympus C5060 camera incorporated into the microscope’s optical system. The samples were photographed using 4× and 10× microscopic lenses. For the purposes of photographic documentation, the samples were illuminated using two fibre illuminator bulbs. For colour correction, a blue conversion filter BC12 (120 mired) was placed behind the lenses to increase the colour temperature. Photographs of the pigmented wood sections at the site of the impregnant application were analysed in terms of the depth of pigment penetration into the structure of the wood. Photographic sample images were scaled (µm/pix) using 4× and 10× lenses to adequate length models. The measured length started at the edge of the sample and ended at the point where the pigment was no longer present. In each sample, photograph measurements were taken in 30 randomly selected locations, after which the obtained results were analysed statistically.

### 2.10. Statistical Analysis of the Results

The statistical analysis was performed using the ANOVA module of Statistica 13.3 software. A multivariate analysis of variance was conducted where the factors (quality predictors) were the type of wood and impregnant used, while the dependent (described) variable was the depth of impregnation. To determine the significance of discrepancies in terms of the penetration depth, Tukey’s HSD test was used (α = 0.05). Before the test, the quality of variances was verified (Levene’s test) to confirm that the variances were homogenous, and that Tukey’s test could be employed.

## 3. Results

Table 1 presents the mean results of surface tension measurements conducted for the hydrophobic liquids. For comparison, the results obtained for distilled water are also included.

The surface tension measured in both analysed hydrophobic liquids was nearly 70% lower than in the case of distilled water. Of the two, the surface tension of the vegetable oil-based liquid was 4.7% higher than that of the mineral-oil based alternative.

Table 2 presents the mean results from the measurement of the contact angle after the application of distilled water onto the surfaces of wood samples (pine, aspen, oak) prior to impregnation and after the absorption and drying (drying time 24 h) of the two impregnants (MOA, VOA). Table 3 presents example photographs of the distilled water drops applied on the wood surfaces during the contact angle measurements.

Of the untreated wood samples, the lowest contact angle values were observed for pine wood (θ = 23.2°). A drop of distilled water could easily spread and wet the pine surface. After the application and drying of the vegetable oil-based hydrophobic liquid, the contact angle increased nearly twofold, while for the mineral oil-based hydrophobic liquid, the value increased by even more than two times. After impregnation, the wettability of pine wood decreased. Unfortunately, due to the high values of standard deviation relative to the average values of the contact angle measurements during the application of distilled water to the surface of aspen and oak, we were only able to identify certain tendencies. In the case of oak, it can be assumed that both VOA and MOA impregnation increase hydrophobicity. On the other hand, in the case of aspen, impregnation with the two analysed agents did not affect the wettability of this wood material.

At the subsequent stage of the study, an attempt was made to evaluate the process of the samples’ aerobic biodegradation. The process entails the biological decomposition of organic matter into simple compounds, mediated by living organisms such as bacteria, fungi, protozoa, algae, actinobacteria, or worms, with the help of natural factors such as oxygen, water, or sunlight. It is a highly complicated process that reduces complex substances into simple compounds that can then be absorbed by plants, hence facilitating the circulation of matter in the ecosystem [29].

Figure 1 presents the results of a 28-day study on the susceptibility of vegetable oil-based hydrophobic liquid (VOA) to ultimate aerobic biodegradation, measured with the method of manometric respirometry in accordance with the OECD 301F guidelines, at a temperature of 22 ± 1 °C. On the 28th day of the experiment, the susceptibility to aerobic biodegradation, expressed as the ratio of BOD/COD, reached 83.4% for the analysed material. Pursuant to the OECD guidelines, the threshold level of high susceptibility to biodegradation measured with the use of respirometry is 60% of the chemical oxygen demand (COD). The threshold was reached within a 10-day window. At the end of the 10-day window, the observed ratio of aerobic biodegradation reached approximately 69.5% for the analysed material.

The results of the corresponding analysis conducted for the other (commercially available) hydrophobic liquid are presented in Figure 2. The 28-day test to establish the susceptibility to the ultimate aerobic degradation of the mineral oil-based liquid (MOA) was conducted with the method of manometric respirometry at 22 ± 1 °C. In this case, on the 28th day of the study, the susceptibility to aerobic biodegradation measured as the ratio of BOD/COD reached 47.8%. Hence, the threshold level of high susceptibility to biodegradation provided in the OECD guidelines for respirometry measurements, i.e., 60% of the chemical oxygen demand (COD), was not reached.

Table 4 presents the results of liquid density and viscosity measurements. Based on the results obtained, it can be observed that VOA was 7.95% less dense than liquid MOA. In terms of dynamic viscosity, the results differed very significantly as the value recorded for VOA was as much as 3.135% lower.

The results of the tests conducted while reducing the water content in the respective wood samples are presented in Table 5. As follows from the table above, the intended result was achieved. The water content in the respective types of material was reduced. In the case of pine wood, the water content decreased by 9.13% relative to the original value. In the case of aspen wood, the decrease was the most substantial, 18.32%, while in oak the content decreased by 9.75%.

Wood density can be easily treated as a marker of its yield, quality, and durability, which determines the physical, mechanical, and technological properties of timber, particularly its hardness and abrasibility [30,31]. The density of wood depends, among other factors, on the tree species, water content, part of the tree it was harvested from, tree growth conditions, and the presence of defects. By definition, wood density is the ratio of its mass to volume at a given moisture level or when dry [32]. As follows from the conducted measurements (Table 6), certain discrepancies in terms of density were observed for the respective timber types. In the case of pine, its dry density was 390 kg/m^3^, i.e., 2.5% less than the initial value. The result was consistent with the ranges reported by Kotwica et al. (2004) [33]. The density of dried aspen wood was 610 kg/m^3^ and was 11.59% lower than the initial measurement. The density of dried oak wood was 475 kg/m^3^, i.e., 9.52% less as compared to the value obtained before drying the wood sample.

The absorbability of wood indicates the capacity of wood submerged in water to absorb the liquid, and depends mostly on the type of wood and duration of submersion. Completely dry wood has the highest capacity to absorb water as the liquid can penetrate cellular membranes, maximising the amount of free and bound water absorbed into the material (PN-D-04119:1959 [34]). The process is particularly significant in the context of structural timber and wood used in construction [35]. The water content in wood is constantly fluctuating and affects its mechanical properties. This problem has already been considered in numerous studies conducted worldwide [36,37]. All strength properties of the material are reduced due to water absorption by hygroscopic wood, which also makes it easier for microorganisms to attack the material [38]. The amount of water that can be absorbed by wood depends on its density and diffusion coefficients, which determine the rate at which water can travel inside the material.

Table 7 and Table 8 present the results in terms of absorbability analyses for untreated wood and wood treated with the respective hydrophobic liquids. As indicated by the experimental results, the samples were characterised by fast sorption in the initial phase of soaking, after which the absorbability gradually decreased at the later stages of the experiment. The initial rate of sorption was due to capillary uptake, during which the liquid migrated and propagated via the capillary tubes, vessels, and cellular walls of the wood material [37].

The results of the experiments conducted on raw material indicated that within 6 h, oak and aspen wood reached considerably lower levels of absorbability than pine wood. Significant differences in terms of wood absorbability relative to the control samples were observed for pine wood at 83.33% (10.00%), while the values for aspen and oak were, respectively, 33.19% (10.80%) and 41.77% (22.40%).

The phenomenon can be explained by the presence of natural capillary tubes in wood, which quickly reach equilibrium in the hydrative medium by inhibiting capillary flow [39].

Table 8 presents the rate of water absorption by the wood samples after the application of the two analysed preparations within 6 h.

When samples coated with the protective hydrophobic liquids underwent the absorbability analysis, a sharp initial increase in the parameter was observed. Despite the impregnation, the mass of the wood samples increased after their submersion in water; however, the increase was considerably less significant than in the case of raw wood without the protective coating. After the VOA treatment, the mass of the particular wood samples increased by, respectively, 26.65, 30.76, and 30.08%. For MOA, the mass of the wood samples increased by, respectively, 33.12, 36.84, and 50.21%. The observed discrepancy in terms of the mass of samples by wood type was, respectively, 24.26% for pine, 19.73% for aspen, and 66.90% for oak, in favour of VOA.

After the application of VOA, the absolute water content after 6 h of soaking was, respectively, 38.23% for pine wood, 85.69% for aspen wood, and 62.11% for oak wood. After the designated period of time, the initial absorption rates in the series were, respectively, 6.40, 10.20, and 11.10%/h. After the use of MOA, the absolute water contents of the respective samples were 34.71, 105.43, and 61.67%. After the designated time of the analysis, the initial absorption rates in the series were, respectively, 5.80, 14.50, and 9.10%/h.

### Spectroscopic Analysis of the Materials—FTIR

Figure 3 presents the infrared spectra measured for the analysed samples. Table 9 presents all the characteristic bands observed, and identifies the corresponding functional group vibrations. In the spectra obtained from the VOA and control samples, there is a clear indication of a characteristic spectral band skeleton typically found in various triglyceride fractions.

The data available in the literature facilitate an increase in accuracy in associating the vibrations of the respective functional groups with specific corresponding bands, both in natural and synthetic samples, but the task nonetheless remains far from trivial in materials such as those analysed in this study [40,41,42,43,44,45,46,47,48,49]. The aim of the spectroscopic analysis presented below was to provide in-depth molecular characteristics of the selected products, which could then be relatively easily related to their quality. Furthermore, we also hoped to identify specific marker bands that would greatly facilitate such analyses.

Starting from the highest wavenumber side, we should first mention the characteristic =C-H stretching vibrations (in the trans- transformation) with the maximum at ~3500 cm^−1^ (Figure 3), which originate from the vibrations of triglyceride fractions [41]. The band was very poorly visible in the VOA and control samples, while, in the case of the MOA sample, it was overlapped by a band corresponding to the stretching vibrations of -OH hydroxyl groups with the maximum at approx. ~3346 cm^−1^, which was not observed in the other two samples. Next, the vibrations with the maximum at ~2917 and 2841 cm^−1^ originated from the –C-H stretching vibrations in the -CH_3_, CH_2_ groups found in the aliphatic groups of triglycerides [40,46,47,48,49]. Next, in both the VOA and control samples, we observed a band with the maximum at ~1743 cm^−1^, which corresponded to the stretching C=O vibrations [50]. The band was not present in the MOA sample. A widening of the band was observed on the lower wavenumber side where a slight enhancement with the maximum at ~1710 cm^−1^ was present, possibly originating from the hydrogen-bound carbonyl groups mainly found in acidic groupings, likely in the C=O…H-O- conformation [51]. The band with the maximum at ~1640 cm^−1^ most likely originated from -OH deformation vibrations. Slightly before that, at approximately 1620 cm^−1^, we observed the stretching vibrations of cis-transformed C=C groups; in the other samples, we observed the stretching vibrations of C=C groups in the cis-transformation [46]. For MOA, we observed stretching vibrations at 1548 cm^−1^, characteristic of the C=C group, which were not present in the other samples. Next, at ~1460 cm^−1^, there was a very characteristic area with a fairly rich selection of bands. The said maximum corresponds to the deformation vibrations of the -C-H groups in -CH_2_ and -CH3 groupings, primarily scissor deformation vibrations. Only the VOA sample returned a band with the maximum at ~1420cm^−1^, originating from the deformation vibrations of the -CH_2_ groups present in the structure of fatty acids. Next, at ~1240 cm^−1^, we registered the important maximum of a band characteristic of -C-H vibrations. Notably, the vibrations of the ester bond ν(C-O) are composed of two combined, asymmetric vibrations, primarily of the C-C(=O)-O and O-C-C groups [50]. Characteristically for oily substances of such origin, the former vibration tends to be more intensive [46]. In the case of the analysed samples, primarily VOA and the control, unsaturated esters produced very intensive vibrations with the maxima at ~1162, 1092, and ~1030 cm^−1^, characteristic of C-O and C-O-C stretching vibrations and associated with primary alcohols [40]. Both types of esters are present in triglyceride molecules. In turn, the band with the maximum at ~1240 cm^−1^ is often associated exclusively with the out-of-plane bending vibrations of methylene groups [52]. The band with the maximum at ~904 cm^−1^, which was present in all of the analysed samples, originated from the stretching vibrations of cis-substituted olefin groups [41]).

Vibrations in the region below 900 cm^−1^ corresponded to the out-of-plane deformation vibrations of -HC=CH- groups in the cis- conformation, as well as the wagging vibrations of the same groups (δ(-(CH_2_)n- and -HC=CH- (*cis*-)) [40,46]. In the MOA sample, we clearly observed the lack of practically any vibrations in the 1350–1150 cm^−1^ region, which evidenced the absence or significantly smaller presence of the corresponding structures in the analysed product.

By comparing the FTIR spectra registered for the VOA, MOA, and control, we could observe clear and important discrepancies in a number of spectral ranges. The differences were due primarily to the different origins, i.e., natural or artificial, of the particular samples. The most evident discrepancies were observed in the following regions: 3346, 1743, 1640, 1460, and the entire region from 1200 to 1000 cm^−1^ with the maximum at 1162 cm^−1^. The band with the maximum at 3346 cm^−1^ was registered only in the MOA sample, indicating a significant presence of water molecules, which may negatively impact the durability of the protective treatment itself as well as the material treated. In the case of the other two samples, we observed practically no bands in this particular region. This discrepancy was further underscored by the band with the maximum at 1640 cm^−1^. Next, bands originating from carbonyl groups were observed in the VOA and control oils, but were not present in MOA, which may have a bearing on product durability. Furthermore, particularly interesting differences were observed in the range below 1480 cm^−1^, where the spectra registered for the control and VOA samples returned considerably richer spectral results. Consequently, it can be concluded that the samples contained considerably more structures of this type, as is characteristic of oily samples (namely of C=C, C-O, and C-H groups).

As evidenced by the results of the presented FTIR measurements, a molecular analysis provides a fast and reliable source of information on a given product’s quality. On the one hand, we can observe that the natural sample was characterized by a considerable prevalence of bands indicating the presence of various fatty acid fractions, which can have a direct bearing on the quality and effectiveness of the product. On the other hand, we also saw evidence that the artificial sample contained considerable quantities of water molecules, which may be a negative indication for the types of uses considered herein. This seems to corroborate the validity of the research direction chosen. At the same time, the results also suggest that spectroscopic analysis may indeed serve as a fast, cheap, and reliable preliminary method of analysing product quality. It is our intention to continue exploring this research direction in our future work.

Drops of stained water were placed on the surfaces of wood samples coated with the vegetable oil-based and mineral oil-based hydrophobic liquids to see whether they would remain on the surface of the material (Figure 4, Figure 5 and Figure 6).

Based on the above images, it can be concluded that only in the case of the aspen samples (Figure 5b) the effect of water spreading on the wood surface was successfully prevented with the vegetable oil-based impregnant. In the remaining samples, both in the control and the VOA samples, evident spreading of the stained water drop can be observed. On the other hand, the use of the mineral oil-based agent (Figure 4c, Figure 5c and Figure 6c) succeeded in preventing the spreading of the stained water drop, regardless of the type of wood.

In the subsequent part of the study, the depth of liquid penetration into the wood structure was analysed microscopically (Figure 7).

The analysis of the impregnation penetration depth conducted with the use of Tukey’s test in the ANOVA module allowed the identification of two homogenous groups (Table 10). As revealed in the analysis, the type of agent used during impregnation significantly influenced the depth of the liquid’s penetration into the wood structure (Figure 4, Figure 5 and Figure 6 and Table 7). Overall, the mineral oil-based impregnant was able to penetrate deeper into all of the wood samples, as compared to its vegetable oil-based counterpart. In the case of oak and pine, the respective penetration depth was 110% greater on average, and in the case of aspen it was 50% greater.

As follows from Washburn’s equation, a liquid with a dynamic viscosity *η* and surface tension *γ* will penetrate a capillary for which the pore radius is *r* to the depth *L*, as per the following formula:(6)$$L=\sqrt{\frac{\gamma rtcos(\phi )}{2\eta }}$$
where

*ϕ*—contact angle between the penetrating liquid and the solid.

Given the similar values of the surface tension of the analysed impregnates (VOA—27.6 mN/m; MOA—26.3 mN/m), it can be assumed that the penetration depth is, in this case, strongly dependent on the liquid viscosity. Significantly higher MOA viscosity allowed this agent to penetrate deeper into the wood structure of each analysed wood sample.

What also follows from the results of the analysis is that the depth of impregnant penetration is highly dependent on the structural differences between the respective materials. The structure of oak is naturally dense, which inhibits impregnant penetration (Figure 6). Hence, the results recorded for this type of wood were the lowest after both the VOA and MOA treatments. The liquid penetration observed in pine and aspen samples was considerably deeper. Due to the more porous structure of these wood types, the depth of penetration recorded in their case was, on average, approximately 100% greater than in the case of oak after the application of VOA. Notably, the penetration levels observed for this impregnant also varied statistically significantly between pine and aspen samples themselves (Table 10). In all the analysed wood samples, the penetration depth achieved by MOA also revealed statistically significant discrepancies (Table 10). The lowest values (approx. 69 µm) were recorded for the oak samples, and the highest (approx. 135 µm) for the pine samples.

## 4. Conclusions

The use of natural oils in various industries has helped us to reduce the adverse environmental impacts of modern agriculture. The conducted biodegradability analyses indicated that the use of vegetable oils in the production of hydrophobic liquids is consistent with the general turn towards more environmentally friendly solutions in industry. As follows from the analysis of aerobic biodegradability, the vegetable oil-based alternative demonstrated 83.4% susceptibility, while for the mineral oil-based hydrophobic liquid that value was only 47.8%, which confirmed the considerably lesser biodegradability of the latter liquid.

It was demonstrated that the use of hydrophobic liquids reduced wood absorbability as compared to corresponding results obtained for unimpregnated wood, and the resulting level of absorbability depended on the type of wood and the protective treatment applied. The observed discrepancies between wood samples coated with the MOA and VOA liquids were, respectively, 24.26% for pine wood, 19.73% for aspen, and 66.90% for oak, in each case in favour of the vegetable oil-based liquid.

The study demonstrated that the depth of wood penetration was considerably higher for the MOA impregnant, as compared to the VOA alternative. The lower penetration capacity of VOA means that an overall lesser quantity thereof will be needed to impregnate a given wood surface compared to the MOA product. As such, the higher penetration capacity of MOA will also likely translate to a greater cost of wood treatment and a longer time needed for the impregnant coat to fully dry.

The FTIR analysis conducted on the samples revealed significant spectral discrepancies in regions associated with the content of water, as well as clear differences in those associated with the fatty acids present in the natural samples. The molecular discrepancies evidenced the high quality of the proposed products, and therefore their viability as alternatives to the commercial products currently available on the market.

## Figures and Tables

**Figure 1 materials-16-04975-f001:**
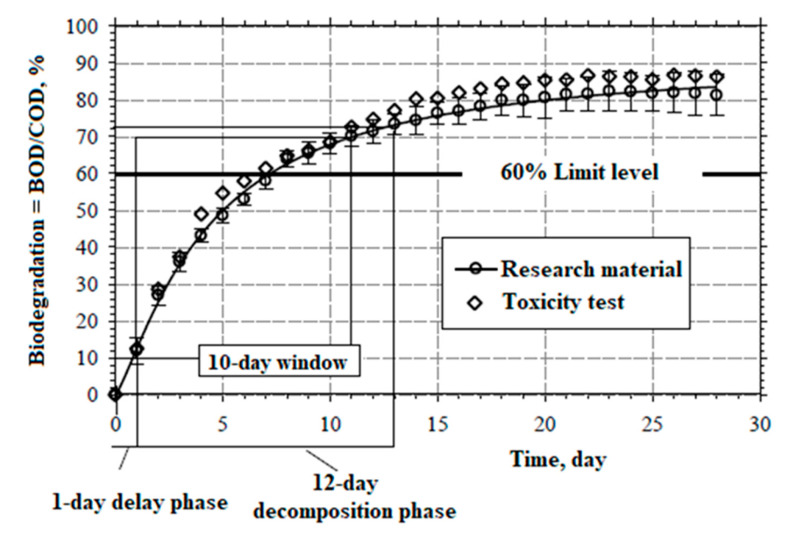
Process of biodegradation and toxicity control for the vegetable oil-based liquid.

**Figure 2 materials-16-04975-f002:**
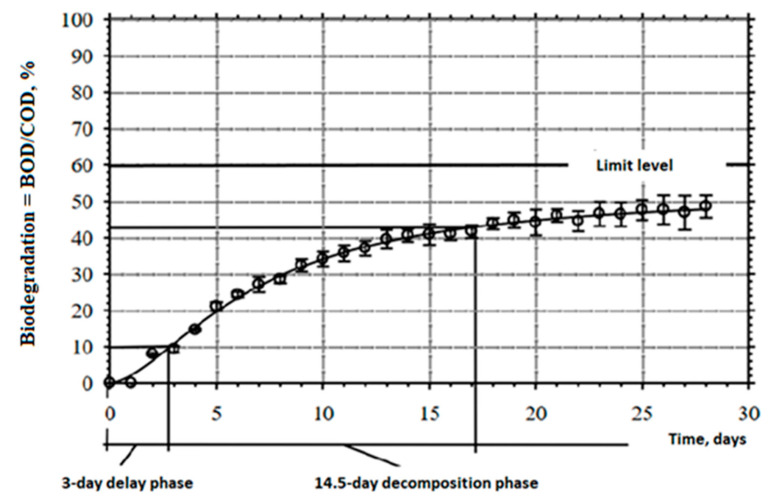
Process of biodegradation and toxicity control for the mineral oil-based liquid.

**Figure 3 materials-16-04975-f003:**
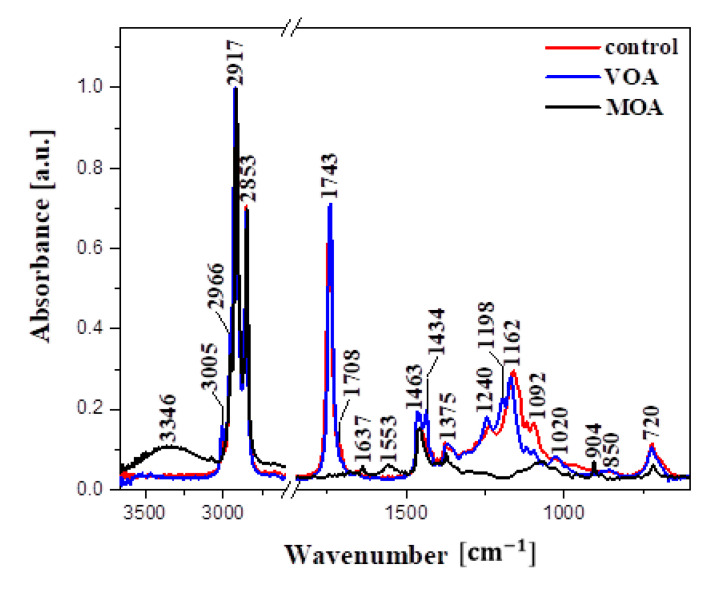
FTIR spectra for the analysed oil samples: control, VOA, and MOA, registered in the spectral range from 400 to 3700 cm^−1^.

**Figure 4 materials-16-04975-f004:**
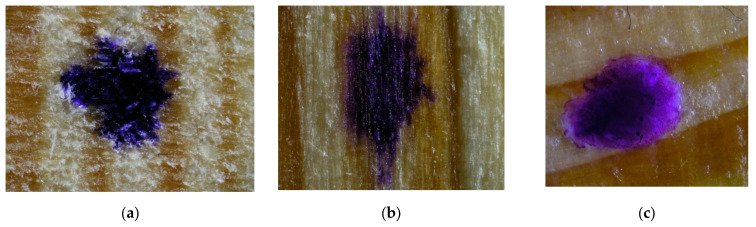
Stained water drops on the surface of pine samples, ×20 magnification. (**a**) control, (**b**) sample impregnated with VOA, (**c**) sample impregnated with MOA.

**Figure 5 materials-16-04975-f005:**
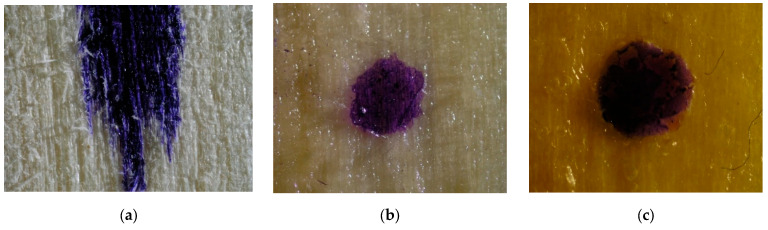
Stained water drops on the surface of aspen samples, ×20 magnification. (**a**) control, (**b**) sample impregnated with VOA, (**c**) sample impregnated with MOA.

**Figure 6 materials-16-04975-f006:**
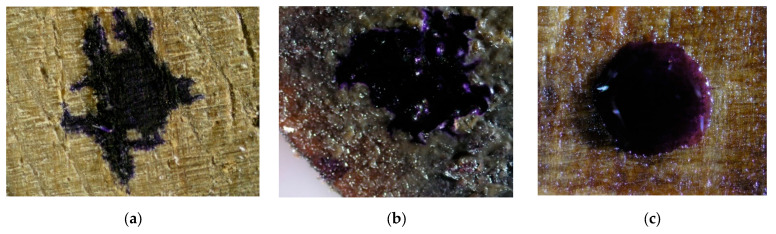
Stained water drops on the surface of oak samples, ×20 magnification. (**a**) control, (**b**) sample impregnated with VOA, (**c**) sample impregnated with MOA.

**Figure 7 materials-16-04975-f007:**
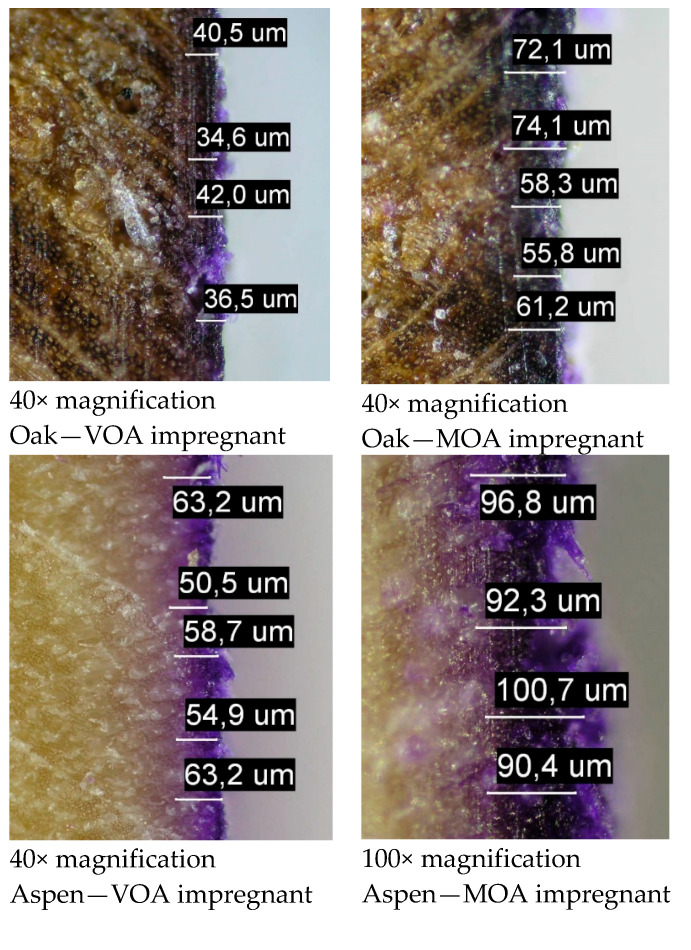
Penetration depth of the analysed impregnants for the respective types of wood; photographs with measurement overlays.

**Table 1 materials-16-04975-t001:** Mean results of surface tension measurements for hydrophobic liquids and distilled water.

Liquid	Surface Tension
	[mN/m]
Distilled water	72.0 ± 1.02
VOA	27.6 ± 0.51
MOA	26.3 ± 1.49

**Table 2 materials-16-04975-t002:** Mean contact angle values after the application of distilled water onto the wood sample surfaces prior to and after impregnation using the two hydrophobic agents.

Type of Wood	Contact Angle		
	[°]		
	No Impregnation	With Impregnation	
		VOA	MOA
Pine	23.2 ± 8.43	45.8 ± 6.79	50.8 ± 6.68
Aspen	54.9 ± 8.36	44.1 ± 4.35	53.7 ± 10.79
Oak	40.1 ± 19.03	55.4 ± 3.80	59.9 ± 5.03

**Table 3 materials-16-04975-t003:** Example images of drops photographed during contact angle measurements after the application of distilled water onto the wood sample surfaces prior to and after impregnation with the two hydrophobic agents.

Type of Wood	Contact Angle θ		
	[°]		
	No Impregnation	After Impregnation	
		VOA	MOA
Pine	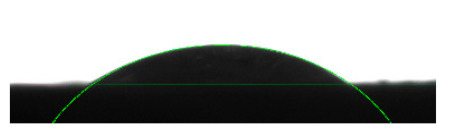	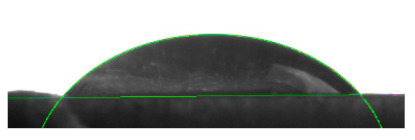	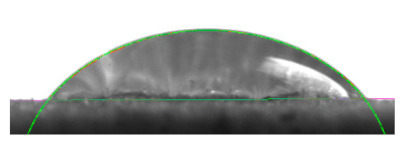
Aspen	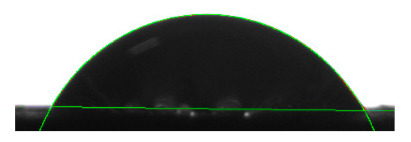	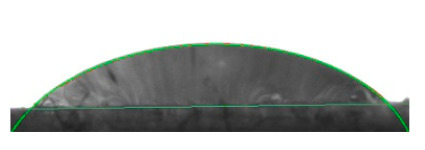	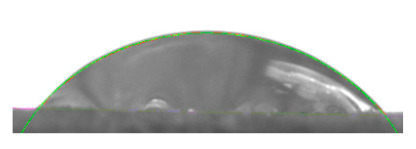
Oak	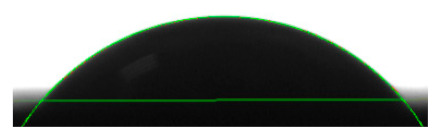	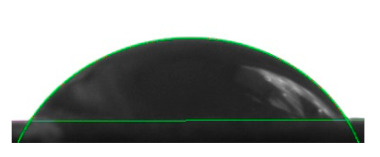	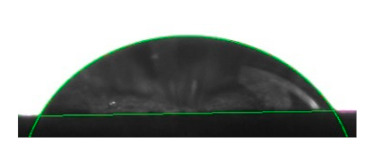

**Table 4 materials-16-04975-t004:** Results of liquid density and viscosity measurements.

Type of Hydrophobic Liquid	Density	Dynamic Viscosity
	[g/cm^3^]	[m·Pa]
Vegetable oil-based liquid	0.88 ± 0.02	7.3 ± 0.03
Mineral oil-based liquid	0.95 ± 0.5	236.19 ± 1.07

**Table 5 materials-16-04975-t005:** Measurements of water content in the respective wood samples.

Type of Wood	Natural Mass of the Sample	Dried Mass of the Sample	Relative Water Content of the Sample
	[g]		[%]
pine	69.08 ± 7.12	62.77 ± 6.43	10.03 ± 0.06
aspen	96.99 ± 1.12	79.22 ± 1.19	22.43 ± 2.02
oak	76.47 ± 3.69	69.02 ± 3.7	10.80 ± 0.20

**Table 6 materials-16-04975-t006:** Wood density analysis.

Type of Wood	Density for the Water Content [W]	Water Content During the Measurement	Density at 12% Water Content	Dry Wood Density
	[kg/m^3^]	[%]	[kg/m^3^]	[kg/m^3^]
Pine	405 ± 0.22	15.00 ± 0.02	400 ± 1.32	390 ± 2.33
Aspen	695 ± 0.24	24.00 ± 0.01	690 ± 1.54	610 ± 2.45
Oak	530 ± 0.43	14.00 ± 0.02	525 ± 1.33	475 ± 2.42

**Table 7 materials-16-04975-t007:** Measurements of raw wood absorbability.

Type of Wood	Mass of Dry Wood	Mass after 6h of Soaking	Relative Water Content after 6h of Soaking W	Initial Absorption Rate in the Vn Series
	[g]		[%]	[%/h]
Pine	3.24 ± 0.29 ^a^	5.94 ± 0.12 ^b^	10.00 ± 0.61	13.9 ± 0.80
Aspen	4.85 ± 0.10 ^a^	6.46 ± 0.20 ^b^	22.40 ± 0.11	5.50 ± 0.09
Oak	4.62 ± 0.44 ^a^	6.55 ± 0.24 ^b^	10.80 ± 0.80	10.52 ± 3.75

Values designated by different small letters in the columns of the table are significantly different (α = 0.05).

**Table 8 materials-16-04975-t008:** Analysis of wood absorbability after the application of antiadhesive/hydrophobic compounds.

Type of Wood	Mass of Dry Wood	Mass after 6h of Soaking	Relative Water Content after 6h of Soaking W	Initial Soaking Rate	Initial Soaking Rate in the Series
	[g]		[%]	[%/h]	[%/h]
**MOA**					
Pine	4.68 ± 1.42 ^a^	6.23 ± 1.32 ^b^	38.23 ± 1.58	6.36 ± 0.24	6.40 ± 0.26
Aspen	4.56 ± 2.32 ^a^	6.24 ± 4.12 ^b^	85.69 ± 17.30	14.30 ± 3.06	10.20 ± 3.08
Oak	4.64 ± 10.12 ^a^	6.97 ± 11.64 ^b^	62.11 ± 26.16	11.77 ± 5.40	11.10 ± 5.55
**VOA**					
Pine	4.69 ± 0.12 ^a^	5.94 ± 0.21 ^b^	34.71 ± 1.50	5.80 ± 0.24	5.80 ± 0.34
Aspen	4.55 ± 0.21 ^a^	5.95 ± 2.32 ^b^	105.43 ± 5.54	17.59 ± 0.94	14.50 ± 0.96
Oak	4.62 ± 10.12 ^a^	6.01 ± 12.32 ^b^	61.67 ± 19.85	10.27 ± 3.33	9.10 ± 3.45

Values designated by different small letters in the columns of the table are significantly different (α = 0.05).

**Table 9 materials-16-04975-t009:** The location of the maxima of FTIR absorption bands, with assignment of particular vibrations to the respective samples: control, VOA, and MOA [40,41,42,43,44,45,46,47,48,49].

FTIR			Type and Origin of Vibrations
Positioning of Band [cm^−1^]			
CONTROL	VOA	MOA	
-	-	3346	ν(-OH)
3007	3007	-	ν_m_(=C-H, *cis*-)
2951	2951	2951	ν_as. vst_(-C-H_vst_, -CH_2_) and ν_s. vst_(-C-H, -CH_2_) (aliphatic groups in triglycerides)
2923	2923	2917	
2853	2853	2848	
1743	1740	-	ν_vst_(-C=O) in esters/ν_vw_(-C=O) in acids
1712	1701	-	
1652	1654	1675/1637	ν_vw_(-C=C-, *cis-*)
1556	1560	1554	δ_vw_(-C-H) in CH_2_ and CH_3_ groups. deformation (scissor)/ ν_vw_(-C-H, *cis-*) deformation (wagging)
1463	1463	1459	
1440	1435	-	
1417	1420	-	
1402	1400	1400	
1377	1366	1373	ν_w. m. vw_(-C-H, -CH_3_) and deformation
1318	1318	1320	δ_m_(-C-H, -CH_3_)
1302	1302	1296	ν_m_(-C-O) or δ_m_(-CH_2_-)
1278	1278	-	
1239	1245	1250	
1160	1194/1170	-	ν_st_(-C-O) or δ_st_(-CH_2_-)
1142	-	1143	
1119	1119	-	
1095	1095	1092	
1060	-	1069	
1029	1029	1029	
-	987	987	
963	-	961	δ_w_(-HC=CH-, trans-) *out-of-plane* deformation
912	915	904	
874	883	883	
857	857	854	
-	844	830	
-	806	808	
-	784	784	
768	769	763	
-	-	742	
722	722	717	

ν—stretching vibrations. δ—deformation vibrations. s—symmetric. as—asymmetric. st—strong. w—weak.

**Table 10 materials-16-04975-t010:** Comparison of wood penetration depth in Tukey’s test (α= 0.05).

Type of Wood	Impregnant	Penetration Depth [µm]	Depth ± Std. Dev.
PINE	VOA	66.6 ^a^	66.6 ± 4.4
	MOA	135.5 ^b^	135.5 ± 10.5
ASPEN	VOA	61.5 ^a^	61.5 ± 6.5
	MOA	96.4 ^b^	96.4 ± 5.9
OAK	VOA	31.9 ^a^	32.0 ± 5.5
	MOA	69.1 ^b^	69.0 ± 5.7

Values designated by different small letters in the columns of the table are significantly different (α = 0.05).

## Data Availability

Not applicable.

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
