# Peer review of "Comparative Analysis of Vegetable and Mineral Oil-Based Antiadhesive/Hydrophobic Liquids and Their Impact on Wood Properties"

_materials, 2023, doi:10.3390/ma16144975_

Round 1

Reviewer 1 Report

The manuscript is interesting. Revision is required before publication.

1. Discuss in detail the density, and viscosity measurement.

2. In Fig.7, the absorption peaks need to be mentioned in detail with references.

3. Explain the role of penetration depth in the current work.

4. The measured absorbability is not having suitable clarity. Rewrite it.

5. What is novelty of this work?

6. Compare the present work with literature, and provide the best.

7. Why to choose this work?

no

Author Response

Dear Reviewer,

Thank you very much for all your comments and suggestions. We have attached the answer file. We hope that our answers will be sufficient and meet your expectations...

We thank you in advance and apologize for any inconvenience.

Yours faithfully,

Marta Krajewska

Reviewer 2 Report

1- Pge one line 18-32: Abstract need to improve because of it didnt containe any excellent results 

2- page 2,3 line 94-109: the aime of the work is not clear to understand , so it need toi rewrite to be easly understand 

3- page 7 line 564: in  figure 7 what is the meaning of AOR &AOM

4-pge 18 line 602: Conclusions  is so long, it need to summarize 

5- page 19 line 644: References need to update

 Minor editing of English language required

Author Response

(The authors gave the same response as above.)

Reviewer 3 Report

The paper is generally well-written, however, revisions are needed.

Title

Wood Propertiese – should be properties

Introduction

The Intro is well-written

What is the hypothesis of your study?

Line 105-109 – it is inappropriate to put it in Intro, maybe Conclusion would be more suitable.

Materials and method

Line 112-115 – can you provide justification why these materials were selected?

Line 116 – please italicized scientific name

Line 185-187 – please revise the sentence

Results and discussion

Please rearrange the subsection so that the sequence match to that of Materials and method, or vice versa.

Line 291-292: “Of the two, the surface tension of the vegetable oil-based liquid was 4.7% lower than that of the mineral-oil based alternative.” But in Table 1, VOA had higher surface tension, please check.

Please revise Table 2

Can Figure 1 and Figure 2 combine?

Line 356 – “Based on the same…” what do you mean?

Line 359 – “3,135%” is it necessary to compare this way?

Line 364 – “ moisture content” or water content, please use consistently.

Table 9, please check again the grouping of Tukey’ test, it is weird.

Figure 6  - please indicate the direction of wood

Spectroscopic analysis of the materials – FTIR – superscript cm-1 please.

There is no discussion on the results. The author should provide more in depth discussion especially on the why caused the difference between two antiadhesive as well as wood species.

The conclusion is OK.

Author Response

(The authors gave the same response as above.)

Reviewer 4 Report

I have really plenty of comments and suggestions. Please see the attached file!

The quality of English written language needs moderate improvement.

Author Response

(The authors gave the same response as above.)

Reviewer 5 Report

This paper has novelty and is informative to the readers. However, considerable changes are required to improve the quality of the manusript.

1. Add a brief sentence to the abstract and point out the implication of this research.

2. In the introduction, explain the knowledge gap explicitly and elaborate on how the outcome of this research could bridge this gap.

3. Please add some literature that used wood/tree-based materials as eco-friendly preservatives. One example could be:

Dong, H., Bahmani, M., Rahimi, S., & Humar, M. (2020). Influence of copper and biopolymer/Saqez resin on the properties of poplar wood. Forests11(6), 667.

4. Please compare the results of this study with previous relevant studies and justify any accordance or contradiction.

5. Conclusion part is too long. It also resembles the abstract. Please summarize it and focus on the key points and gist of the manuscript (takeaway message).

Please avoid complex and elongated sentences. Use active voice and declarative sentences. Review the punctuation throughout the entire text.

Author Response

(The authors gave the same response as above.)

Round 2

Reviewer 3 Report

I appreciate the hard works by the authors in revising the manuscript. The whole manuscript has been greatly improved and it is now acceptable at its current form.

Reviewer 4 Report

Thank you very much for all explanations and the changes you have performed in the text. Now, I do not have any additional remarks.

The quality of English written language is better as in the first version. I am not sure, but it still needs minor editing.

Reviewer 5 Report

The authors satisfactorily addressed the comments and made the required corrections. Therefore, the revised version of this manuscript is acceptable for publication.